# Extracellular proteolysis of tandemly duplicated pheromone propeptides affords additional complexity to bacterial quorum sensing

**Alonso Felipe-Ruiz[1], Sara Zamora-Caballero[1], Shira Omer Bendori[2], José R. Penadés[3,4], Avigdor Eldar[2], Alberto Marina[1]***

1 Instituto de Biomedicina de Valencia (IBV)-CSIC and CIBER de Enfermedades Raras (CIBERER)-ISCIII, Valencia, Spain, 2 Shmunis School of Biomedicine and Cancer Research, Faculty of Life Sciences, Tel-Aviv University, Tel-Aviv, Israel, 3 Centre for Bacterial Resistance Biology, Imperial College London, United Kingdom, 4 School of Health Sciences, Universidad CEU Cardenal Herrera, CEU Universities, Alfara del Patriarca, Spain

* amarina@ibv.csic.es

**Data Availability Statement:** Structures described in the manuscript are available at PDB (http://www.

## Abstract

Bacterial interactions are vital for adapting to changing environments, with quorum sensing (QS) systems playing a central role in coordinating behaviors through small signaling molecules. The RRNPPA family is the prevalent QS systems in *Bacillota* and mediating communication through secreted oligopeptides, which are processed into active pheromones by extracellular proteases. Notably, in several cases the propeptides show the presence of multiple putative pheromones within their sequences, which has been proposed as a mechanism to diversify peptide-receptor specificity and potentially facilitate new functions. However, neither the processes governing the maturation of propeptides containing multiple pheromones, nor their functional significance has been evaluated. Here, using 2 Rap systems from bacteriophages infecting *Bacillus subtilis* that exhibit different types of pheromone duplication in their propeptides, we investigate the maturation process and the molecular and functional activities of the produced pheromones. Our results reveal that distinct maturation processes generate multiple mature pheromones, which bind to receptors with varying affinities but produce identical structural and biological responses. These findings add additional layers in the complexity of QS communication and regulation, opening new possibilities for microbial social behaviors, highlighting the intricate nature of bacterial interactions and adaptation.

## Introduction

Bacterial interactions are a dynamic and intricate process crucial for orchestrating adaptive responses to changing environments. In this complex interplay, quorum sensing (QS) cell to cell communication systems play a pivotal role [1], enabling bacteria to gauge population

pdb.org) with accession numbers 8RST, 8RSU, 8RTC, 8RTE and 8RSV. All peptides found during homology search, thermofluor and thermophoresis data generated during and/or analysed during the current study are available in the provided Supporting Information files.

**Funding:** This work was supported by grants PID2019-108541GB-I00 and PID2022-137201NB-I00 from Spanish Government (Ministerio de Ciencia e Innovación), PROMETEO/2020/012 by Valencian Government and the European Commission NextGenerationEU fund (EU 2020/2094), through CSIC's Global Health Platform (PTI Salud Global) to A.M, by grant MR/X020223/1 from the Medical Research Council (UK), and grant EP/X026671/1 from the Engineering and Physical Sciences Research Council (EPSRC, UK) to J.R.P; by the Israel Science Foundation grant No. 2288/2021 to A.E and grant ERC-2023-SyG Proposal n° 101118890 TalkingPhages (from EU) to A.E., A.M. and J.R.P. A.F-R. received a FPU predoctoral fellowship from Spanish Ministry of Universities, reference FPU19/00433. The funders had no role in study design, data collection and analysis, decision to publish, or preparation of the manuscript.

**Competing interests:** The authors have declared that no competing interests exist.

**Abbreviations:** dTm, denaturation temperature; HPLC, high-pressure liquid chromatography; MGE, mobile genetics element; MST, microscale thermophoresis; OPP, oligopeptide permease; QS, quorum sensing; SEC-MALS, size exclusion chromatography coupled with multi-angle light scattering; SP, signal peptidase.

density and synchronize behaviors through small signaling molecules. Communication by QS systems involves the production and excretion of signaling molecules or pheromones that accumulated in the bacterial environment and are sensed by specialized receptors [2].

Proteins of the RRNPPA family are the most frequent pheromone receptors in *Bacillota* and use small secreted oligopeptides as pheromones [2]. The RRNPPA family received its name from the founding members: Rap from *Bacillus* species, Rgg from *Streptococcus* species, NprR and PlcR from the *Bacillus cereus* group, PrgX from *Enterococcus faecalis*, and AimR from *Bacillus* phages and other mobile genetics elements (MGEs) [3,4]. The RRNPPA family of proteins is extremely important in the biology and virulence of different species and their MGEs, being involved in cell differentiation, including sporulation, biofilm formation and competence development (Rap [5,6] and Rgg [7] systems), control of the phage life cycle (AimR [8]), gene transfer (PrgX [9]), or necrotrophism (NprR and PlcR [10]) between others.

Mature RRNPPA pheromones are genomically encoded peptides of 5 to 10 amino acids, codified immediately downstream or upstream of the RRNPPA receptor gene in form of a pro-peptide of 50 to 120 amino acids. Propeptides are exported and secreted by the SEC pathway and then matured by several proteases [11]. The maturation process is initiated by the signal peptidase (SP) that removes the N-terminal leader peptide and continued by other extracellular proteases, producing the active version of the pheromone. Typically, the mature peptide that acts as active pheromone is located at the C-terminal end of the propeptide, but pheromones located internally in the propeptide have also been shown to work [12]. Once the mature pheromone is produced, it reinternalizes using the oligopeptide permease (OPP) and bounds to its receptor. Receptors usually show a high specificity for mature pheromones produced from their own propeptides [13]. Pheromone binding to their cognate receptor produces a conformational change affecting the disposition of its N-terminal portion, which mediates the receptor binding to the DNA or protein target [14]. Thus, the interactions between the cognate peptides and their receptors modify cellular behavior and gene expression [4,14]. In AimR and Rap systems, pheromone recognition usually inhibits the binding capacity of the receptor to its target, although most members of the RRNPPA family the pheromone induces the opposite effect, promoting its activity [15].

While most members of the RRNPPA family act as transcriptional factors through DNA binding, Rap receptors are specialized members that exhibit the capability to develop their function by interaction with specific protein targets. The common targets of Rap proteins are ComA and Spo0F, 2 master response regulators in the processes of competence and sporulation, respectively [16,17]. However, other Rap-target proteins have been identified, such as the response regulator DegU or specific repressors of MGEs [18–20]. Binding of the pheromone produced from the propeptide, which is named Phr in this RRNPPA subfamily, induces Rap receptor inhibition. Remarkably, Rap receptors are frequently found on MGEs, which regulate MGE- or host-specific processes [18], adding an additional layer of intricacy to their regulatory roles.

Interestingly, propeptides sometimes includes multiple identical or slightly modified sequences of what seems to be the mature pheromones. The number of these repetitions can vary between 1 and 4 [13]. It has been proposed that these peptides with multiple putative pheromones were generated by duplication in a process that could facilitate the diversification of peptide-receptor specificity. In this way, the system keeps one copy of the original pheromone while allowing the others to diverge and gain novel functions [13]. Additional biological functions underlying pheromone duplication have not been discarded [21].

However, despite the abundance of duplications across different RRNPPA systems, whether various mature peptides are produced, how they are generated, and the functions they may have (if any) remain unresolved. Another intriguing question is why, in some cases, the

putative pheromones present in the propeptide have identical sequences, while in other cases, they differ. Here, we examine and verify the hypothesis that some propeptides encode not one but several functional mature pheromones. Our results show that specific maturation mechanisms are required to produce these multiple mature pheromones and that, in the case of pheromones with different sequences, these bind to their cognate receptor protein with different affinities. Our findings introduce an extra layer of complexity to QS regulation in RRNPPA family that enable new avenues for more complex social behaviors.

## Results

### Interactions of Rap^3T protein with putative pheromones

In recent years, *Bacillus* phage phi3T has garnered attention in the realm of bacterial communication. This interest stems from a pivotal study in 2017 [8], which unveiled the presence of a Quorum Sensing (QS) system (AimR^phi3T). This system was shown to regulate the lysis-lysogeny decisions within bacteriophage populations, dependent on their density [8]. However, further exploration revealed the existence of a secondary QS system within phi3T, belonging to the Rap family [22]. To understand the maturation process of RRNPPA systems that presumably encode more than a single putative mature pheromone within their signaling propeptide, we analyzed the Rap system present in the phi3T phage. This system, composed of the protein receptor (Rap^3T) followed by a propeptide (Phr^3T) that shows a pseudo-repetition that offers 2 alternative potential candidates for the mature QS pheromone. The first candidate corresponds to the canonical C-terminal peptide of 6 residues with the sequence SRGHTS and a second potential candidate would correspond to an internal pseudo-repetition of 6 residues, RRGHTA, separated by 14 amino acids (Fig 1A). These 2 putative mature peptides only differ in the amino acids at their ends (positions 1 S/R and 6 S/A), maintaining the conserved RGHT central part of the pheromone (Fig 1A). We have used here the term canonical for the C-terminal peptide since in the majority of Phr propeptides there is a single C-terminally coded mature pheromone and the mechanisms of its maturation have been established [23,24].

To assess the binding capability of these putative pheromones to Rap^3T, a thermal shift assays was carried out. In these assays, the binding of the peptide to the protein results in a conformational change that compact the receptor and increases its denaturation temperature [14]. In addition to the 2 putative pheromones RRGHTA and SRGHTS, we included in the analysis an extended (RRGHTAS) and a shorter (RGHTS) version of the internal and C-terminal putative pheromones, respectively, that could represent alternative forms of their maturation. All peptides stabilized the receptor by more than 20°C, although differences of up to 12 degrees in the denaturation temperatures (dTm) were observed between the different peptides (Fig 1B and 1D). Stabilization occurs upon peptide binding as confirmed by the peptide ERPVGT, the pheromone belonging to the Rap-Phr system of phage phi105 and used here as negative control, that did not show any impact on the stability of Rap^3T (Fig 1B). Interestingly, the internal pheromone RRGHTA showed the highest stabilization with a dTm of 75°C while the canonical C-terminal hexapeptide pheromone showed a much lower (63°C) dTM (Fig 1B and 1D). The shorter and longer version of the internal and C-terminal pheromones respectively, showed lower dTms than the corresponding 6 residues versions (Fig 1B and 1D), confirming that the hexapeptides are most likely the functional ones. These results suggest that the 2 putative pheromones seem to be functional but present different binding affinities or induce different conformational changes in Rap^3T.

Next, we used microscale thermophoresis (MST) to calculate the binding affinity of these pheromones to Rap^3T. MST experiments showed a clear correlation with thermal shift assays corresponding higher dTm with stronger peptide affinity. All 4 peptides bind to Rap^3T in the

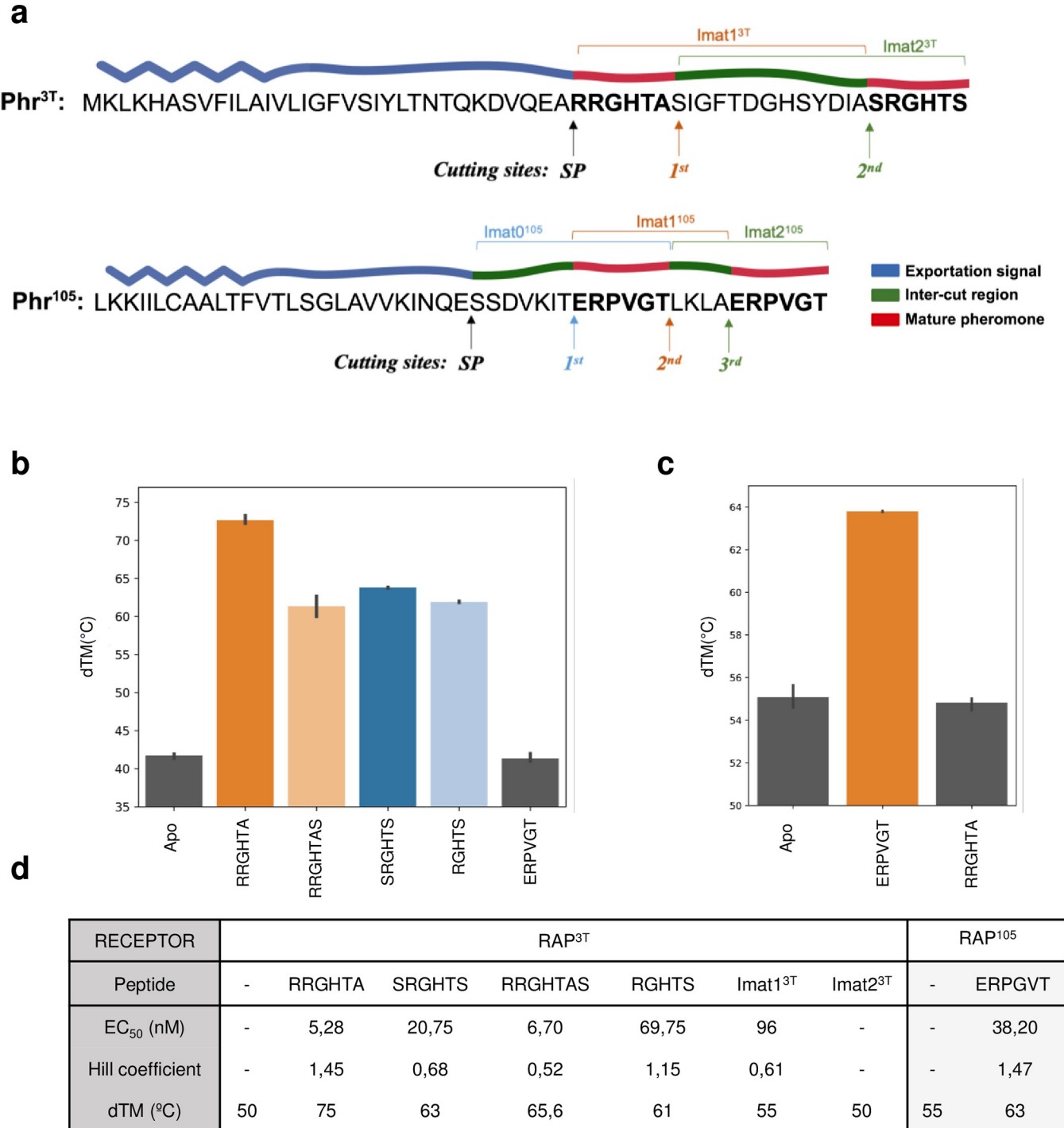

**Fig 1. Characterization of Phr[3T] and Phr[105] propeptides.** (**a**) Sequence of Phr[3T] (top) and Phr[105] (bottom) highlighting the putative mature pheromones in red, the proteolytic cutting site for complete maturation and the exportation sequence (blue). The synthetic peptides used to analyze the propeptide maturation are indicated on the top of the corresponding Phr. (**b**) Denaturalization temperature (dTM) computed in the thermal shift assays for Rap[3T] in the apo form and in presence of the putative pheromones and control. The raw thermal shift assays data can be found in S1 Data. (**c**) dTM in the thermal shift assays for Rap[105] in the apo form and in presence of the putative pheromone and control. Error bars mark standard errors. The raw thermal shift assays data can be found in S2 Data. (**d**) Results of the MST and thermal shift assays on pheromone binding for Rap[3T] and Rap[105]. The raw MST data can be found in S3 Data. dTm, denaturation temperature; MST, microscale thermophoresis.

low nM range, but the internal Phr$^{3T}$ pheromone has the highest affinity (EC$_{50}$ = 5,28 nM). Only a 20% reduction in the affinity (EC$_{50}$ = 6,70 nM) was observed for the possible maturation variant of the internal Phr$^{3T}$ pheromone (RRGHTAS) with an extra amino acid (Figs 1D and S1). Rap$^{3T}$ shows 4 times lower affinity (EC$_{50}$ 20,75 nM) for the putative Phr$^{3T}$ C-terminal canonical pheromone (SRGHTS) and more than 10 times lower affinity (EC$_{50}$ 69,75 nM) for the possible maturation variant of this peptide with one less amino acid (Figs 1D and S1). Interestingly, and in agreement with the dimeric organization observed in the crystal structures of Rap proteins [14,15,25], the thermophoresis assay showed cooperative binding with Hill coefficients that exceeded 1 for the Phr$^{3T}$ internal pheromone and lower than 1 for the C-terminal one (Fig 1D), indicating positive and negative peptide-binding cooperativity, respectively. These results indicate that the internal peptide of Phr$^{3T}$ would be the most active pheromone once matured but also that Rap$^{3T}$ is a receptor which can also sense differentially the second pheromone produced from the system.

Additionally, we calculated the binding affinities for 2 intermediate peptides, which we hypothesized could be produced as intermediates during the process of propeptide maturation. These peptides contained, in addition to the 6 pheromone residues, the inter-pheromone region of 14 aminoacids. One of them, Imat1$^{3T}$ corresponds to Phr$^{3T}$ after being processed by the signal peptidase (SP cutting site) and the protease that releases the C-terminal pheromone (second cutting site), and the second one, Imat2$^{3T}$, encompasses the C-terminal pheromone plus the 14 amino acids of the inter pheromone region produced by a putative protease that cleaves the internal pheromone at its C-terminal end (first cutting site) (Fig 1A). Notably, Rap$^{3T}$ could bind to Imat1$^{3T}$ with weak affinity but not Imat2$^{3T}$ (Figs 1D and S1), supporting that N-terminal elongation of the active pheromone is a major determinant in peptide recognition.

## Phr$^{3T}$ maturation

While the previous results indicated a greater affinity of the internal peptide for the Rap protein compared to the canonical one, it remained unclear whether the mature forms of the internal and the C-terminal encoded peptides are indeed produced. To clarify this point, we undertook a comprehensive study of Phr$^{3T}$ maturation, employing bioinformatic and experimental approaches. Computational analysis with SignalP 5.0 software [26] identified the initial proteolytic cleavage mediated by signal peptidase (SP cutting site), likely linked to propeptide export, as previously described in Rap-Phr systems [21,27]. This cleavage therefore generates a shorter peptide of 24 residues containing the 2 putative pheromones under study, which will require the action of additional proteases with at least 2 additional proteolytic cuts to generate the mature forms of these peptides (Fig 1A).

To identify these proteases and confirm the existence of mature peptides, we capitalized on the robust stabilization exhibited by mature pheromones on Rap$^{3T}$ in thermal shift assays reported in the previous section. We employed Imat1$^{3T}$ and Imat2$^{3T}$, the 2 putative immature peptides including the pheromones (Fig 1A), and these peptides underwent incubation with supernatants from *B. subtilis* 168 strain, which we anticipated would contain the extracellular proteases necessary for the maturation of the Phr$^{3T}$ propeptide. Subsequent to incubation, thermal shift assays were conducted with Rap$^{3T}$ guided by the following rationale: although Imat1$^{3T}$ and Imat2$^{3T}$ peptides exhibited significantly impaired binding and stabilization of Rap$^{3T}$ in thermal shift assays (Fig 2A), we hypothesized that if mature peptides were generated, they will stabilize the Rap protein, indicating their existence. This outcome would also implicate certain unknown proteases in the generation of these peptides.

We found that *B. subtilis* 168 supernatants effectively processed both propeptides as denaturation temperatures observed for Imat1$^{3T}$ and Imat2$^{3T}$ after incubation were identical to

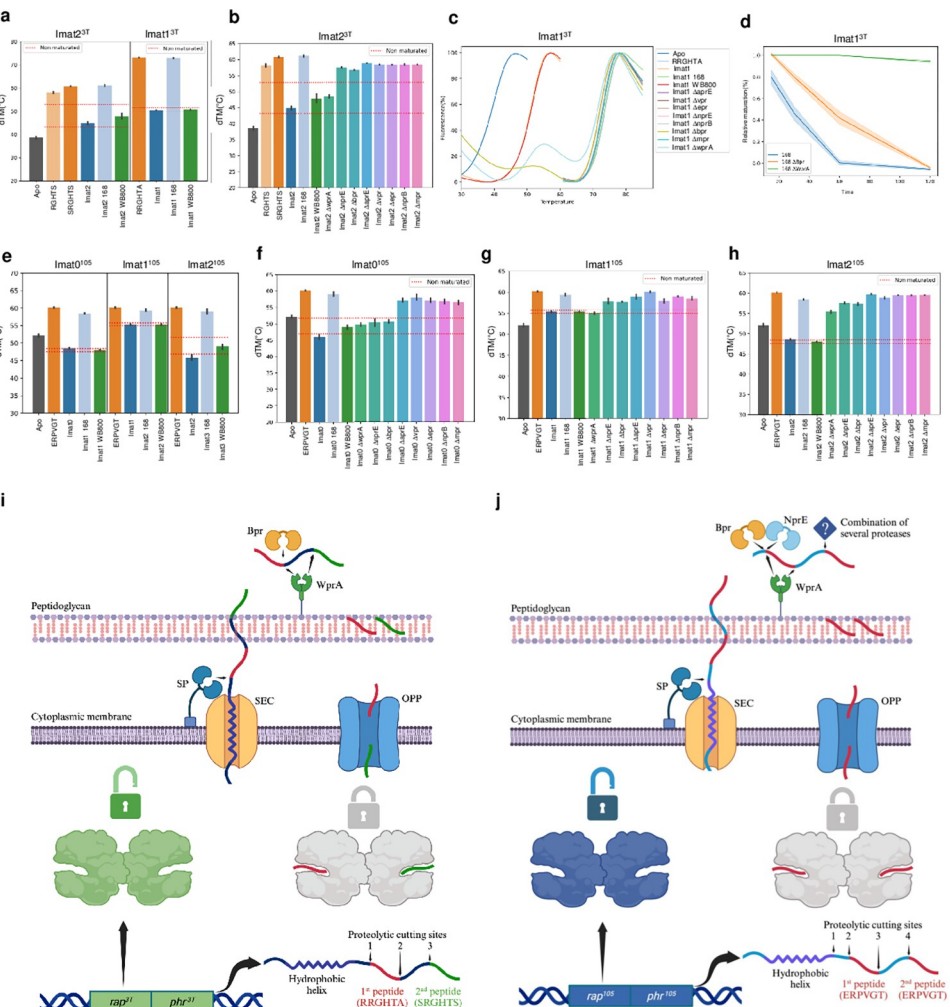

**Fig 2. Propeptide maturation of Phr$^{3T}$ and Phr$^{105}$ propeptides.** Thermal shift assay experiments for maturation effect of supernatants from *B. subtilis* 168 and WB800 strains in immature version of the pheromones codified in Phr$^{3T}$ (**a**) and Phr$^{105}$ (**e**), mean values of triplicates are plotted, 3 standard deviations from negative control maturation (WB800 supernatant) are highlighted in red. Error bars mark from the minimum to maximum values observed in the triplicates. Thermal shift assays with supernatants from *B. subtilis* 168, WB800, and individual protease mutant strains in maturation of peptides Imat2$^{3T}$(**b**), Imat1$^{105}$ (**f**), Imat2$^{105}$ (**g**) and Imat3$^{105}$ (**h**), mean values of triplicates are plotted, 3 standard deviations from negative control maturation (WB800 supernatant) are highlighted in red. Error bars mark from the minimum to maximum values observed in the triplicates. (**c**) Thermal shift slopes of the effect of individual mutations of 8 protease genes on Imat1$^{3T}$ peptide maturation. (**d**) Kinetic assays comparing Imat1$^{3T}$ peptide maturation in 168 supernatant and supernatants of WprA and Bpr protease KO mutants, relative maturation was calculated as fluorescence of the peak corresponding with the mature pheromone divided by the fluorescence of the peak corresponding with the non-maturated pheromone. (**i**) Schematic representation of Phr$^{3T}$ maturation: signal peptidase initiates first proteolytic cut after propeptide exportation, and proteolytic cuts 2, and 3 mediated by Bpr and/ or WprA. (**j**) Schematic representation of Phr$^{105}$ maturation: signal peptidase initiates first proteolytic cut after propeptide exportation, and proteolytic cuts 2, 3, and 4 mediated by a combination of proteases. Fig 1I and 1J are created with Biorender.com. The data underlying this figure can be found in S4–S6 Data. OPP, oligopeptide permease; SP, signal peptidase.

those produced by the corresponding internal and C-terminal mature pheromones, respectively (Fig 2A). As a control, these immature peptides were incubated with supernatants of *B. subtilis WB800*, an extracellular protease-deficient derivative strain mutant in *nprE*, *nprB*, *aprE*, *epr*, *bpr*, *mpr*, *vpr*, and *wrpA* [28]. Importantly, no increase in Rap$^{3T}$ stabilization was

observed (Fig 2A). These results confirmed that (i) both pheromones can be matured from Phr[3T]; and (ii) that one or several of these 8 proteases are involved in Phr[3T] maturation.

To further elucidate the maturation process and pinpoint the specific proteases, we conducted thermal shift assays after incubating the different peptides with supernatants obtained from *B. subtilis 168* strains in which each of the aforementioned genes expressing proteases were individually mutated. For Imat2[3T], only the Δ*wprA* mutant showed null maturation capability (Fig 2B). Notably, for Imat1[3T], the supernatants from the Δ*wprA* and Δ*bpr* mutants showed 2 slopes with TMs corresponding to those observed for the assays with the protease-deficient (WB800) and WT (168) *B. subtilis* strains, indicating an impaired capacity to generate sufficient mature peptide RRGHTA for all the Rap[3T] protein present in the assay (Figs 2C and S2). Time course maturation assays of Imat1[3T] with Δ*wprA* and Δ*bpr* showed that the Δ*bpr* mutant was able to fully mature the peptide at longer incubation times while the Δ*wprA* mutant shows minimal maturation levels even at these times (Fig 2D). These results point to WprA as the protease responsible for the maturation of both peptides, with Bpr participating less efficiently in the C-terminal cleavage of the internal pheromone. A schema of Phr[3T] maturation can be observed in Fig 2I. However, it cannot be ruled out that the differences observed between both proteases for this last cleavage are due to different levels of expression between them. These findings contrast with previous studies suggesting that CSF and PhrA propeptides from *B. subtilis* 168 are processed by Epr or Vpr proteases [23] and suggest a new layer of regulation in Rap communication, wherein different peptides are matured by the participation of distinct proteases, potentially generating each pheromone based on the context. For instance, in the case of Rap[3T], Bpr is strictly regulated by DegU [29], a major regulator of *B. subtilis* biofilm formation, possibly affecting QS communication.

## Structural characterization of Rap[3T]-peptide complexes shows a uniform peptide-induced conformation

Once the existence of both peptides was proved, our focus moved to the structural characterization of Rap[3T] in presence of all the different peptide tested. Initially, the structure of Rap[3T] was solved in complex with RRGHTA, the peptide with higher affinity and stabilization. The Rap[3T]-RRGHTA complex crystallized in C222$_1$ space group (S1 Table), showing a monomer in the asymmetric unit, which generates by crystallographic symmetry the prototypical dimer observed in other crystal structures of Rap proteins [14,25,30] (S3 Fig). The structure revealed similar architecture to other Rap receptors, with an N-terminal 3-Helix bundle (3HB) domain (residues 1–69), followed by 7 tetratricopeptide repetitions (residues 95–376) conforming the TPR peptide-binding domain, connected by a linker of 26 amino acids (residues 70–94) (Fig 3A and 3B). The putative dimerization of Rap[3T] is due to the interaction of the 2 TRP domains that hide a large surface of TPR7 from the solvent. This surface is generated by the mutual interaction of the TPR7 and other TPR elements that conform the peptide-binding domain (Figs 3C and S3), explaining the cooperative effect observed in pheromone binding.

To molecularly assess the differences in affinity and stabilization for the alternative peptides analyzed we solved the structures for Rap[3T] in complex with RRGHTAS, SRGHTS, and RGHTS (S1 Table). Remarkably, these structures exhibited nearly identical conformations as confirmed by the low differences in the superimposition of their Cα atoms for either individual monomers or the complete dimers (RMSD 0.36–0.50), dispelling the notion of peptides eliciting divergent conformational shifts (S4 Fig). Notice that the molecular conformational are due to peptide binding and not induced by the crystallization conditions since crystals were obtained in 2 different space groups and cell dimensions. Rap complexes with RRGHTA and SRGHTS peptides displayed C222$_1$ space group and with cell dimensions that only

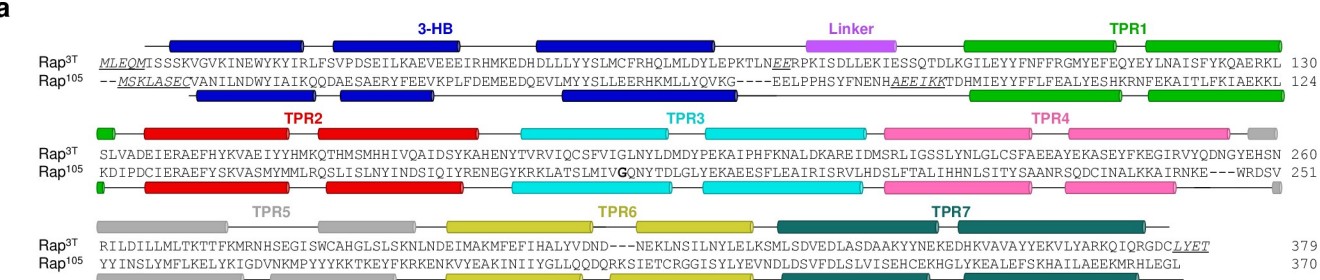

**a**

Rap³ᵀ    *MLEQM*ISSSKVGVKINEWYKYIRLFSVPDSEILKAEVEEEIRHMKEDHDLLLYYSLMCFRHQLMLDYLEPKTLN*EE*RPKISDLLEKIESSQTDLKGILEYYFNFFRGMYEFEQYEYLNAISFYKQAERKL    130
Rap¹⁰⁵   --*MSKLASEC*VANILNDWYIAIKQQDAESAERYFEEVKPLFDEMEEDQEVLMYYSLLEERHKMLLYQVKG----EELPPHSYFNENH*AEEIKK*TDHMIEYYFFLFEALYESHKRNFEKAITLFKIAEKKL    124

Rap³ᵀ    SLVADEIERAEFHYKVAEIYYHMKQTHMSMHHIVQAIDSYKAHENYTVRVIQCSFVIGLNYLDMDYPEKAIPHFKNALDKAREIDMSRLIGSSLYNLGLCSFAEEAYEKASEYFKEGIRVYQDNGYEHSN    260
Rap¹⁰⁵   KDIPDCIERAEFYSKVASMYMMLRQSLISLNYINDSIQIYRENEGYKRKLATSLMIV**G**QNYTDLGLYEKAEESFLEAIRISRVLHDSLFTALIHHNLSITYSAANRSQDCINALKKAIRNKE---WRDSV    251

Rap³ᵀ    RILDILLMLTKTTTFKMRNHSEGISWCAHGLSLSKNLNDEIMAKMFEFIHALYVDND---NEKLNSILNYLELKSMLSDVEDLASDAAKYYNEKEDHKVAVAYYEKVLYARKQIQRGDC*LYET*    379
Rap¹⁰⁵   YYINSLYMFLKELYKIGDVNKMPYYYKKTKEYFKRKENKVYEAKINIIYGLLQQDQRKSIETCRGGISYLYEVNDLDSVFDLSLVISEHCEKHGLYKEALEFSKHAILAEEKMRHLEGL    370

**b**

3-Helix Bundle

Tetratricopeptide repeats (TPR)

**c**

**d**

3-Helix Bundle

Tetratricopeptide repeats (TPR)

**e**

**Fig 3. Structural characterization of Rap-pheromone complexes.** (**a**) Pairwise structural alignment of Rap³ᵀ-RRGHTA and Rap¹⁰⁵-ERPGVT. Structural elements are shown above and below of Rap³ᵀ and Rap¹⁰⁵ sequences, labeling the 3HB domain (blue), linker (purple), and TPR repetitions (various colors). Sequences in italics and underline correspond to disordered regions of the structure. Cartoon representation of the Rap³ᵀ-RRGHTA monomer (**b**) and dimer (**c**) with the structural elements colored as in **a**. In dimer representation, the RRGHTA peptide is showed in sticks with carbon atoms in orange. Cartoon representation of the Rap¹⁰⁵-ERPVGT monomer (**d**) and dimer (**e**) with the structural elements colored as in **a**. In dimer representation, the ERPVGT peptide is showed in sticks with carbon atoms in orange.

allowed the presence of 1 monomer in the asymmetric unit generating the dimer by crystallographic symmetry. Meanwhile complexes with RRGHTAS and RGHTS peptides exhibited P2$_1$ space group and a Rap dimer in the asymmetric unit (S1 Table). Dimeric organization of Rap$^{3T}$ in solution, both in the apo form and bound to the internal and C-terminal Phr$^{3T}$ pheromones was confirmed by SEC-MALS (S5 Fig).

As could be anticipated since all the structures show identical conformation, the analysis the Rap$^{3T}$-peptide complex revealed a quite similar peptide recognition for all of them, with a core amino acid set critical for binding across all peptide versions (Fig 4A and S2 Table). This core comprised N226, a conserved asparagine residue in Rap proteins that anchors the main-chain peptide [14] and up to 23 other amino acids interacting with both pheromone side- and main-chain (Fig 4A and S2 Table). Among this set of residues highlight the role of tyrosines Y151 and Y225 with several contacts with peptide residues in positions from 1 to 5, or D338 and Q182 that interact with the N- and C-terminal peptide ends (Figs 4A and 4B and S6 and S2 Table). It is precisely at these ends where the peptides show binding differences, and the structures show how the receptor adjusts the binding to recognize all peptides. Comparison of complexes with RRGHTA and SRGHTS peptides showed that the sidechain of Arg in position 1 for RRGHTA mediates a network of salt-bridges with D264 and D335 plus hydrophobic interactions with M304 meanwhile the Ser in this position for SRGHTS only makes a hydrogen bound with Y225 (Fig 4B and S2 Table). These differences in interactions for position 1 may explain the lower stabilization seen in thermal shift assays and the higher EC$_{50}$ value from thermophoresis assays for SRGHTS. Accordingly, the complex with the shorter peptide (RGHTS) showed that this position is empty (S6 Fig), explaining why this peptide shows the lowest affinity for the receptor. In contrast, at the C-terminal end, the presence of a Ser in the SRGHTS peptide does not excessively broaden the interactions of Ala in the RRGHTA peptide. The complex shows that the alanine residue presents alternative orientations in both peptides, mimicking the carboxyl-terminal group of the RRGHTA peptide some of the Ser interactions of SRGHTS peptide (Figs 4B and S6 and S2 Tables). Interestingly, the structure of the complex with RRGHTAS peptide showed that the additional Ser residue lacks electron density that prevented its building (S6 Fig), supporting the idea that this residue does not participate in the binding and that RRGHTA is the genuine mature peptide version.

## Rap$^{3T}$ represses both sporulation and competence and is regulated by mature and immature peptides

Multiple Rap receptors target either the response regulator ComA, preventing competence induction, or dephosphorylate Spo0F~P, preventing the phosphorylation of Spo0A and induction of sporulation [15,16]. Previous work has shown that the receptor Rap$^{BA3}$, which shows a 76% of homology with Rap$^{3T}$, represses both sporulation and competence upon overexpression and that this repression is relieved upon addition of the peptide RRGQT [13]. We therefore checked whether overexpression of Rap$^{3T}$ also represses competence and sporulation using the same approach used in that work [13]. We introduced a competence (PsrfA-3xYFP) or a sporulation (PspoIIG-3xYFP) reporter into either the wild-type strain or into a strain carrying the overexpression of $rap^{3T}$ under an inducible promoter (see Methods). We found that expression of both reporters was markedly reduced in the presence of overexpressed Rap$^{3T}$ under appropriate conditions (see Methods), suggesting that Rap$^{3T}$ indeed controls both processes (Fig 5).

Next, we studied the impact of addition of the RRGHTA and SRGHTS putative pheromones to the growth medium where reporter expression was studied. We found that addition

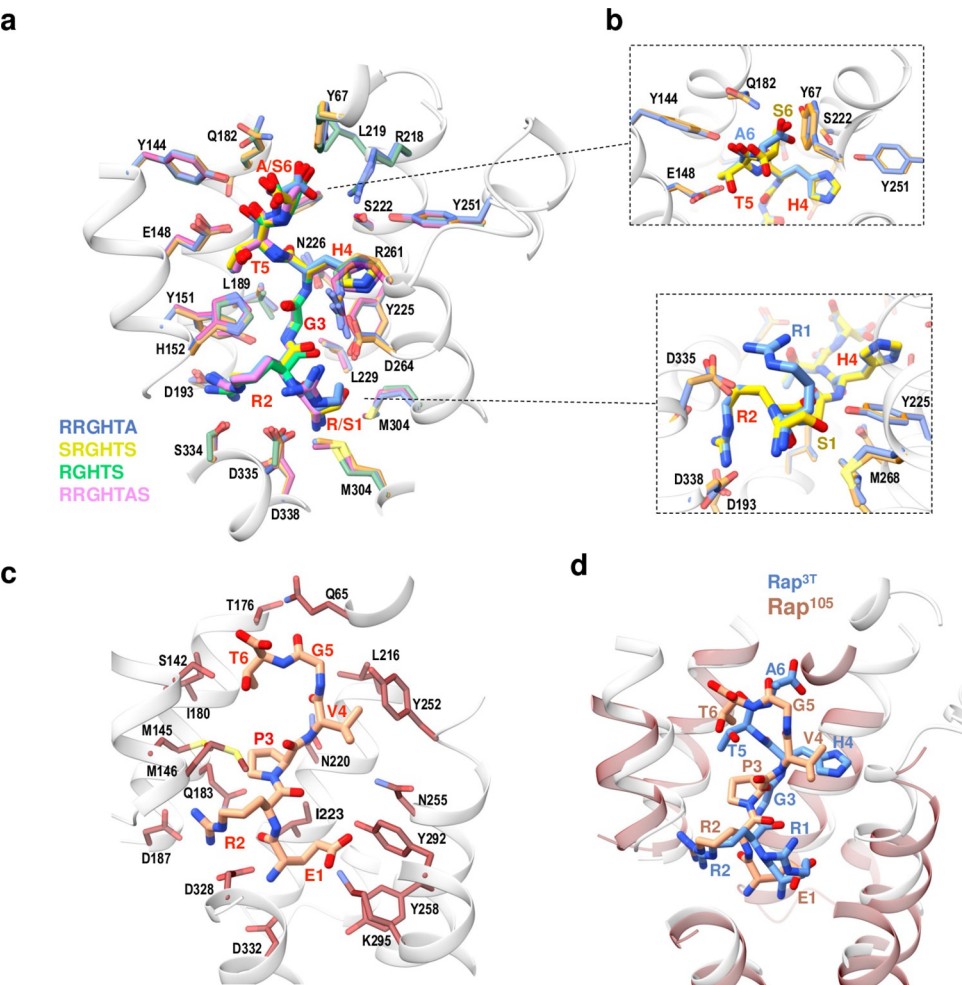

**Fig 4. Pheromones binding and recognition.** (**a**) Pheromone-binding sites from the superposed structures of Rap³T in complex with RRGHTA (blue), SRGHTS (yellow), RGHTS (green), and RRGHTAS (pink) peptides. The peptides and the residues that recognize them in the corresponding structures are shown on sticks with a similar color tone and labeled. Nitrogen, oxygen, and sulfur atoms are shown in navy blue, red, and yellow, respectively. The structural elements where the recognition residues are placed is shown in translucent white cartoon for the Rap³T-RRGHTA structure. (**b**) A close view of the C- (up) and N-terminal (down) ends of the RRGHTA and SRGHTS pheromones and the interacting residues in sticks and colored as in **a** show recognition differences in these positions. (**c**) Pheromone-binding sites from the Rap¹⁰⁵-ERPVGT structure with the pheromone in stick and carbon, nitrogen and oxygen atoms colored in salmon, navy blue and red, respectively. The peptide recognition residues are shown in sticks with carbon atoms in chocolate and labeled. (**d**) Superimposition of Rap³T-RRGHTA (white) and Rap¹⁰⁵-ERPVGT (chocolate) TPR peptide-binding domains place the corresponding pheromones in almost identical positions. Pheromones are shown in stick with carbon atoms in light blue and salmon for RRGHTA and ERPVGT, respectively.

of 10 μm of any of both peptides to the strain overexpressing *rap³T* led to a marked increase in YFP expression. Therefore, both mature peptides are potent for interaction with Rap³T and its inhibition in vivo. Assays with the control peptide ERPVGT confirms that the effect is specific to both pheromones. Finally, we used Imat1³T and Imat2³T peptides to check in vivo the pro-peptide maturation. The addition at the same concentrations of both immature peptides produced similar levels than the mature pheromones of reporter expression (Fig 5), confirming their in vivo cleavage into the mature functional peptides.

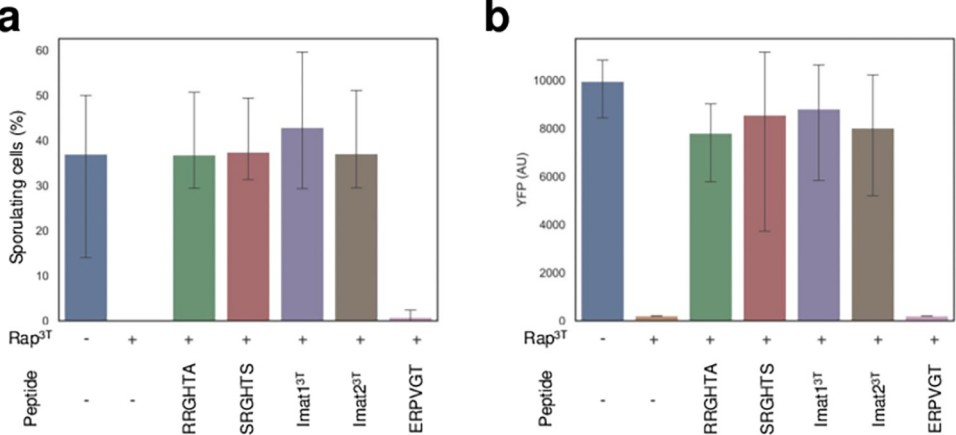

**Fig 5. Sporulation and competence assays for the Rap³ᵀ system.** (a) Percentage of spores obtained in sporulation assays after overexpression of the *rap³ᵀ* gene in the presence or absence of different regulatory peptides. (b) YFP levels of a competence reporter assay integrated into strains overexpressing Rap³ᵀ in the presence or absence of different regulatory peptides. Error bars mark standard errors. T-Student statistics were computed for all conditions compared to the negative control (empty vector). Significance is indicated as "ns" ($p$-value >0.05), and $p$-values are displayed when statistical significance is observed. The average value of triplicates is presented, with whiskers indicating the range from the minimum to maximum values observed in the triplicates. The data underlying this figure can be found in S7 Data.

## Characterization of Phr¹⁰⁵ maturation with an alternative model of pheromone duplication

Next, we investigated another system employing propeptide duplication as a potential additional regulatory layer in QS signaling. The Rap-Phr system from phage phi105 caught our attention as its Phr (Phr¹⁰⁵) carried 2 identical copies of the putative mature pheromone (ERPVGT). One copy corresponds to the canonical pheromone located in the C-terminal region of the propeptide and the second is located internally separated by a spacer of only 4 amino acids (LKLA) (Fig 1A). This type of Phrs with duplications suggested a different regulatory mechanism by propeptide maturation process that would alter the receptor:pheromone ratio.

Once we identified the presence of 2 identical pheromones in the propeptides, we aimed to understand if both are produced and how they are produced. First, we computational scrutinized the Phr¹⁰⁵ propeptide looking for signal peptidase recognition motifs, revealing a putative signal peptidase cleaving site between E27 and S28, 7 amino acids before the N-terminal end of the putative mature form of the internal pheromone (Fig 1A). This implies that to generate the 2 copies of the pheromone from Phr¹⁰⁵, a minimum of 4 proteolytic cleavages are required; the one produced by the signal peptidase (SP site) during export, those at the N- and C-terminal ends (sites 1 and 2) of the internal peptide and that corresponding to the N-terminal end (site 3) of the canonical terminal peptide (Fig 1A). To test whether these proteolytic cleavages were occurring, we performed a similar approach to that used previously for Phr³ᵀ, but in this case 3 immature synthetic peptides were required. Imat0¹⁰⁵ (SSDVKIT**ERPVGT**) includes the internal pheromone between SP site and site2, Imat1¹⁰⁵ (**ERPVGT**LKLA) the internal pheromone between sites 1 and 3, and Imat2¹⁰⁵ (LKLA**ERPVGT**) the C-terminal pheromone between site 2 and C-terminal end (Fig 1A).

Thermal shift assays showed that while the mature version of Phr¹⁰⁵, ERPVGT, induced a stabilization of 9 degrees in Rap¹⁰⁵, the immature versions promoted minimal stabilization (Imat1¹⁰⁵) or even induced destabilization (Imat0¹⁰⁵ and Imat2¹⁰⁵), indicating that only the mature pheromone binds productively to the receptor (Fig 2E). Upon incubation with *B. subtilis*

*168* supernatants, an enhanced stabilization matching the effect of the mature pheromone was observed for all the immature peptides (Fig 2E), confirming that all the immature peptides can be processed to the final active version by the proteases present in the supernatant. We next confirmed that the different peptides used where processed by an external protease by using supernatants from the extracellular protease defective *B. subtilis WB800* strain, which not produce Rap[105] stabilization (Fig 2E). To identity the specific proteases involved in the process, we made use of the derivative *B. subtilis 168* strains carrying individual deletional in one of the 8 extracellular proteases. Imat0[105] showed reduced or null maturation activity in Δ*nprE*, Δ*bpr*, and Δ*wprA* mutants, Imat1[105] only for Δ*wprA* mutants, while Imat2[105] was maturated for all the supernatants (Fig 2F–2H). These results support an elaborate regulation in the production of the different mature pheromones from Phr[105]. While the canonical C-terminal pheromone can be produced by different proteases, suggesting a fast availability as soon as the peptide is exported, the internal pheromone only is produced by a restricted number of proteases a requires the participation of WprA (schematized in Fig 2J), suggesting a higher temporal control in its production.

## Molecular characterization of Rap[105] interaction with its pheromone

To comprehend the interaction between Rap[105] and its predicted mature pheromone (ERPVGT), an MST analysis was performed revealing an $EC_{50}$ of 34nM with a Hill coefficient of 1.41 (Figs 1D and S1). Positive cooperativity again emerged in the binding, underlining the role of cooperativity in pheromone recognition in Rap systems. Interestingly, the MST assays showed that Rap[105] exhibited weaker affinity for its own pheromone than Rap[3T] presented for either of both internal and canonical (10- and 2-fold, respectively) pheromones present in its Phr[3T] propeptide (Fig 1D), suggesting that it requires high concentrations of pheromone in the medium to induce a response.

To further understand the recognition by Rap[105], we solved the structure of Rap[105]-ERPVGT (Rap-Phr[105]) complex at 2.5 Å resolution. Rap-Phr[105] complex crystallized in spatial group P2$_1$ with 2 Rap molecules per asymmetric forming a dimer with similar organization to the observed in Rap[3T] and other Raps (Figs 3D, 3E, and S3 and S1 Table). The structure of the receptor showed the classical architecture of Rap receptors with N-terminal 3HB domain (residues 1–67) followed by 7 TPRs (residues 89–367) (Fig 3A, 3C and 3D). Analysis of the pheromone's binding pocket unveiled Asn220, corresponding to Asn226 in Rap[3T], as the conserved Rap Asn that anchors the pheromone. A further 16 receptor residues (Q65, S142, M145, M146, T176, I180, Q183, D187, L216, I223, Y252, Y258, Y292, K295, D328, and D332) formed direct contacts with the pheromone (Fig 4C and S2 Table). Comparison of the structures of Rap[105] and Rap[3T] in complex with their peptides gives clues to the lower affinity of Rap[105] for the pheromone (Fig 4D). Pheromones in both receptors are well anchored by their N- and C-terminal ends and present an Arg in position two whose side chain shows a similar interaction with the receptor, mainly through a salt bridge with an Asp residue located at identical position (D187/D193 and D338/D332 for Rap[105] and Rap[3T], respectively) (S2 Table). However, the pheromone Phr[105] shows null or weak interactions through two of its 6 amino acids (Pro3 and Gly5) (S2 Table). Also, the presence of this proline in the middle of the pheromone restricts the conformational freedom of the peptide, while in the Phr[3T] this Pro is a Gly that increases pheromone freedom and mediates interactions with 2 tyrosines (Y151 and Y225) of Rap[3T] receptor (S2 Table).

## Distribution and correlation of Rap[3T]-Phr[3T] and Rap[105]-Phr[105]-related systems

A comprehensive search across a data set of Rap propeptides aimed to identify peptides resembling those encoded by the Rap[3T]-Phr[3T] system (see Materials and methods section) identified

56 Phr propeptides (hereafter referred to as the Phr[3T] family) exhibiting candidate mature peptides with similarities to the 2 putative Phr[3T] active peptides (corresponding with more than 80% identity with the 2 mature pheromones present in Phr[3T]). In agreement with previous observations [13,21], 12,5% of the 56 identified Rap-Phr systems were found in MGEs, 4 were encoded by prophages, 1 by phage-plasmids, and 2 by plasmids (Fig 6A and S3 Table). To evaluate whether duplication of peptides can lead to relaxed selection, we analyze the Phr[3T] family. Relaxed selection can result in divergence of one of the peptides which may lead to signal diversification, driven by selective pressures favoring the emergence of new specificities. Our analysis of the variability in the identified peptides revealed contrasting results. The duplicated active pheromones displayed a much higher conservation within the propeptide than inter-pheromone regions (Fig 6B). Similar conservation to the pheromones in the propeptide is only observe at the N-terminal region corresponding to the starting methionine and lysin duplet, which are required for the exportation by SEC traslocon [31]. These findings support the hypothesis that in this case, an ancient duplication in the propeptides is maintained by selection for retaining both functional pheromone copies as well as the region essential for export.

Propeptides with a single putative mature pheromone were identified for 4 variants (RRGHTA, SRGHTS, IRGHTS, and RRGQTA). Interestingly, all these propeptides do not have the active pheromone at its C-terminal position but show additional extensions of variable length (6–18 residues) on this end. While the duplicated propeptides always showed pseudo-repeats (defined as internal non-identical repetitions of the C-terminal pheromone that in other systems can be found as C-terminal pheromone, not suggesting lack or reduced activity) and in no case, we found an identical internal duplication of the C-terminal peptide (Fig 6A and S3 Table) as has been reported for other Rap system families (see below). Duplicate propeptides featuring the first copy as RRGHTA and the second as SRGHTS similar to Phr[3T] were observed for multiple systems, and additional pseudo-repeated propeptides encoding combinations of RRGHTA with (mainly) RRGQTA or SRGSVI also are present. Notice

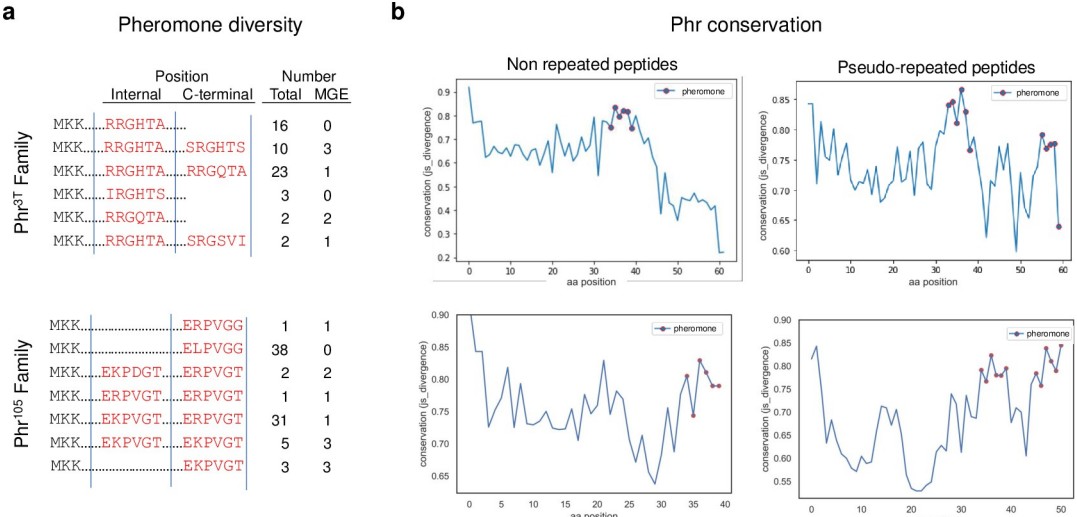

**Fig 6. Conservation patterns of the Phr[3T] and Phr[105] propeptide families.** (**a**) Pheromone architectures and number of receptors in the Phr[3T] (up) and Phr[105] (down) families. (**b**) JS divergence conservation plots generated from non-repeated (left) and pseudo-repeated (right) propeptides of Phr[3T] (up) and Phr[105] (down) families with the residues corresponding to the putative pheromones highlighted as red dots, in Y axis, the calculated Shannon Entropy, in X axis, the position in the alignment. The data underlying this figure can be found in S8 Data. MGE, mobile genetics element.

that RRGHTA always occupies the internal position in the pseudo-repeated peptides (Fig 6A and S3 Table).

In the case of Rap$^{105}$-Phr$^{105}$ system, computational analysis identified 81 Rap-Phr systems (hereafter referred as the Rap$^{105}$-Phr$^{105}$ family) with propeptides exhibiting candidate mature pheromones with high sequence similarity (only 1 or 2 amino acid changes) to Phr$^{105}$ (Fig 6A and S3 Table). Similarly, to Rap$^{3T}$-Phr$^{3T}$, 13,5% of the systems were encoded within MGEs, prophages in this case. Interestingly, almost half of the Phrs of this family have internal identical pheromone repeats (6), as exemplified by Phr$^{105}$, or pseudo-repeats (33) similar to those observed in Phr$^{3T}$ (Fig 6A and S3 Table). Sequence comparison of the Phr$^{105}$ family propeptides showed a similar pattern to Phr$^{3T}$ family with high conservation in the regions corresponding with the mature pheromones, both internal and C-terminal, while the rest of the propeptide remain unconserved (Fig 6B), strengthening the hypothesis that internal repetitions are under strong evolution pressure related with their biological function.

## Discussion

Propeptide duplication has been proposed an evolutionary path to vertical expansion and diversification of Rap-Phr QS systems [13]. In this scenario, similar to the neofunctionalization hypothesis [32], a novel Rap-Phr signaling systems could be generated through divergence of the duplicate pheromone an appropriate coevolution of the receptor. In contrast, duplication and divergence may be maintained without allowing for the evolution of divergence, if each of the duplicates serve a different function. However, this scenario has not been studied. More generally, it has not been analyzed whether these internal duplications are functional during the diversification process. Here, we have focused our investigation on 2 Rap-Phr systems from phages to clarify these questions since they represent 2 different mechanisms of pheromone duplication, with one system carrying identical copies of mature pheromone and in the other are dissimilar. Our findings support a functional role for each of the pheromone in the duplicated pair, restricting the evolutionary path where duplication can lead to divergence in Rap-Phr specificity.

In this context where the different pheromones within the propeptide seem to play a specific function, the maturation process acquires a high relevance. Our results show that this process requires in each case the involvement of different and specific proteases in generating functional pheromones, proposing maturation as an additional layer of regulation. The maturation of propeptides Phr$^{105}$ and Phr$^{3T}$ highlighted the importance of protease WprA in the maturation of duplicated peptides, playing a crucial step in the generation of the mature pheromones. Nevertheless, Bpr and NprE also participate in the production of specific mature pheromones while the C-terminal pheromone of Phr$^{105}$ can be generated unspecifically. Previous studies conducted in PhrA and PhrC postulated Vpr, AprE, and Epr as the proteases implicated in maturation of its canonical pheromones [23], strengthening the notion that different systems may have alternative maturation pathways. In agreement with this idea, the expression of the extracellular proteases is tightly regulated and are expressed under certain conditions also dependent on bacterial population. Bpr transcription is activated under DegU regulon, in biofilm formation conditions [29]. Meanwhile, WprA is expressed under de Sigma-A regulon in conditions of sporulation [33] and the *npr*E transcription is inhibited by the ScoC transcriptional regulator on the exponential phase [34]. Therefore, the production and availability of the proteases introduces and enables additional layers of regulation in the process of peptide maturation and consequently in QS communication. This layer would allow the constant production of the propeptide, which in the case of Rap systems is associated with its receptor but restrict the availability of the mature pheromone to certain metabolic conditions when the specific proteases are produced.

The characterization of the interaction between Rap[105] and Rap[3T] with their cognate pheromones revealed important differences and propose additional alternative functional and regulatory scenarios. Rap[3T] shows much higher affinity for both pheromones of its propeptide than Rap[105] for its unique. The lower affinity shown by Rap[105] could be linked to keep both identical copies after duplication. In this way, Rap[105] activity is highly dependent on the availability of mature pheromone that can be produced from the same Phr propeptide, but this availability depends on an additional regulatory step in the maturation. In this point, we also studied cooperativity in the recognition of the mature pheromone, which was also observed in other receptors of the RRNPPA family [35]. MST experiments provide binding curves that allow us to calculate the Hill coefficient ($>$1 signifies positive cooperativity and $<$1 negative) showing interesting differences in the cooperativity in receptor-peptide binding between receptors and pheromones (Figs 1D and S1). In the case of Rap[3T], the cooperativity differs depending on the recognized pheromone, being positive for the higher affinity internal peptide meanwhile is negative for the C-terminal one of lower affinity. This observation propose a two-speed regulatory scenario where the receptor gives fast response to the availability of the internal pheromone while the response is slower to the presence of the C-terminal one as it requires higher pheromone accumulation. In contrast, Rap[105]-Phr[105] system the receptor has become more specific with a low pheromone affinity, in this context the positive cooperative observed could propose a regulatory scenario where once the high concentration of pheromone required by the receptor is reached, binding to both subunits is accelerated, explaining the presence of 2 identical copies of the mature to increase the receptor:pheromone ratio. Both regulatory scenarios seem to be common since Phr propeptides with several putative pheromone copies varying in only 1 or 2 positions or with multiple (up to 8) copies of the same mature pheromone have been reported [13,21] and observed in our analysis.

In conclusion, this study provides novel insights into the regulatory mechanisms of QS signaling through the Rap-Phr system, particularly in the context of propeptide duplication. We show that the different pheromones present in propeptides with duplications are produced and, more importantly, are active. Furthermore, the production of each mature pheromone is differentially regulated, providing them with function that may select for maintenance of duplication. This direct selective advantage does not contradict the possibility that duplication may lead to divergence, which is a rare evolutionary event. Exploring the maturation process of propeptides, we emphasize the involvement of various proteases in generating functional pheromones. This finding suggests that the maturation process is not uniform across different systems, introducing an additional context-dependent regulatory point to QS signaling communication. The combination propeptide maturation by different proteases regulated by growth conditions and/or the activity of the same QS system, together with the presence of multiple copies of the same or different mature pheromones in the propeptide, expands the spectrum of communication scenarios that RRNPPA systems can generate. Overall, the results broaden our understanding of bacterial communication and underscore the complexity of regulatory mechanisms within the Rap-Phr signaling system, with implications for future studies in the field of bacterial QS and signal transduction.

## Materials and methods

### Identification of the putative receptor-peptide pheromones of Rap[3T]-Phr[3T] and Rap[105]-Phr[105] families

Rap-Phr sequences used in the study were found in previous work [21], briefly, HMM generated from previously described members of the Rap-Phr family was used to search for homologs in a database of curated genomes and accompanied plasmids. Phr propeptides with

                                    

similarities to those carried in Phr$^{3T}$ and Phr$^{105}$ we searched performing BLASTP [36] using the sequence of the mature peptides (RRGHTA, SRGHTS, and ERPVGT, respectively) as query using default parameters for short inputs. Rap-receptor couples corresponding to the selected Phrs were extracted and used in subsequent analysis.

### Conservation analysis

Sequence alignments were performed using MAFFT version 7.471 [37] to perform the alignment with default parameters. Conservation analyses were conducted from sequence alignments using the online server Protein Residue Conservation Prediction, using Shannon Entropy parameter (https://compbio.cs.princeton.edu/conservation/score.html) [38].

### Signal peptidase motif identification

Presence of peptide signals for exportation in propeptides was carried out using SignalP 5.0 software [26] using the option *"Gram-positive bacteria organism group."*

### Bacterial strains

Bacterial strains used in this study are listed in S4 Table. *B. subtilis strains 168 (BGSC [39] 1A1)*, *Δ6 (BGSC 1A1299)*, WB800 [28], BKK14700 (*B. subtilis strains 168$^{ΔnprE}$*), BKK10300 (*B. subtilis strains 168$^{ΔaprE}$*), BKK38400 (*B. subtilis strains 168$^{Δepr}$*), BKK02240 (*B. subtilis strains 168$^{Δmpr}$*), BKK11100 (*B. subtilis strains 168$^{ΔnprB}$*), BKE38090 (*B. subtilis strains 168$^{Δvpr}$*), BKK15300 (*B. subtilis strains 168$^{Δbpr}$*), BKK10770 (*B. subtilis strains 168$^{ΔwprA}$*), 1L11 (ϕ105, accession number NC_048631.1), and 1L26 (ϕ3T, accession number KY030782) were obtained from Bacillus Genetic Stock Centre.

Bacillus strains AES9822 (SacA::PsrfA-3xYFP Cm), AES9824 (SacA::PspoIIG-3xYFP Cm), AES9834 (sacA::Psrf-3xYFP cm amyE::hsRap$^{3T}$ spec), and AES9836 (sacA::Psrf-3xYFP cm amyE::hsRap$^{3T}$ spec) were constructed by natural transformation of the pAEC1003, pAEC1103, and pAEC2836 (see next section) plasmids into a PY79;*xpf* background *Bacillus subtilis* strain.

*Escherichia coli* BL21 CodonPlus (DE3)-RIL strain was obtained from Agilent.

### Plasmids and cloning

Plasmids and primers used in this study are listed in S4 and S5 Tables, respectively. To generate the plasmids pAEC1003 and pAEC1103 [40] were constructed by amplifying the Psrf and PspoIIG promoters using the Psrf-sacA-EcoRI-F/Psrf-sacA-BamHI-R and PspoIIG-EcoRI-F/PspoIIG-NheI-R primer pairs, respectively. The PCR product was digested with EcoRI-HF and BamHI-HF or NheI-HF and cloned into pAEC948 digested with the same enzymes. Plasmid pAEC2836 was constructed by amplifying pAEC1046 using the pDR111-hsRap(3T)-F/pDR111-hsRap(3T)-R primer pair and amplifying the Rap gene from a phi3T lysogen *B. subtilis* genomic DNA using the hsRap3T-gibF/hsRap3T-gibR primer pair. The purified DNA fragments were then ligated using the Gibson assembly protocol.

Rap genes from the phi3T and phi105 phages were amplified using the primers pairs Rap3T lic fwd/Rap3T lic rev and Rap105 lic fwd/Rap105 lic rev and template genomic DNA from Bacillus subtilis 1L26 and a synthetic version of *rap105* gene optimized for *E. coli* expression, respectively (S5 Table). In both cases, additional overhangs necessary for cloning were added (see primers at S5 Table). The purified DNA fragments were then ligated pLicSGC1, which was amplified using the pLIC fwd/pLIC rev primers, using NEBuilder HiFi DNA Assembly (NE Biolabs). The resulting pLIC-Rap$^{3T}$ and pLIC-Rap$^{105}$ plasmids expressed the full-length Rap with an N-terminal 6xHis-tag followed by a TEV protease cleaving site.

 

## Measuring expression by fluorescent reporters

For competence experiments, the expression levels of the srf promoter reporter were determinate in a liquid culture generated from a single colony that was inoculated into MC competence media. The strains were grown at 37˚ with shaking at 220 RPM, with or without addition of 1 mM IPTG. After 4 h of growth, peptides were added at a concentration of 10 μm. Following an additional 1 h of growth, YFP fluorescence was measured using a Beckman Coulter cytoflex flow cytometer.

For sporulation experiments, the expression levels of *spoII*G promoter reporter were determinate in a liquid culture from an overnight starter in LB were diluted 1:100 into Schaeffer's sporulation medium (DSM). The strains were grown at 37˚ with shaking at 220 RPM, with or without addition of 1 mM IPTG. After 4.5 h of growth, peptides were added at a concentration of 10 μm. Following an additional 1.5 h of growth, YFP fluorescence was measured using a Beckman Coulter cytoflex flow cytometer.

## Protein production and purification

Rap$^{3T}$ and Rap$^{105}$ proteins were expressed using *E. coli* strain BL21 (DE3) RIL (Agilent) transformed by electroporation with plasmids pLIC-Rap$^{3T}$ and pLIC-Rap$^{105}$, respectively (S4 Table). A single colony carrying the corresponding expression plasmid was grown overnight at 37˚C in 25 ml of LB medium supplemented with 100 μg/ml of ampicillin and 33 μg/ml of chloramphenicol. The culture was used to inoculate 1L of LB medium (dilution 1:40) supplemented with ampicillin and chloramphenicol and was grown until cells reached an OD at 600 nm of 0.6. Then, the temperature was set to 20˚C and a final concentration of 0.1 mM of IPTG was added. The culture was incubated at 20˚C overnight. Cells were harvested by centrifugation and the pellet was suspended in lysis buffer (100 mM Tris-HCl (pH 8), 250 mM NaCl) and lysed by sonication on ice. Cell debris was removed by centrifugation at 10.000 *g* for 1 h. The supernatant was loaded onto a 1 ml HisTrap FF (GE Healthcare) column equilibrated with lysis buffer, washed with this buffer, and eluted with lysis buffer supplemented with 350 mM imidazole. Fractions containing the purest protein were pooled and digested with TEV protease (25:1 molar ratio protein:TEV) and dialyzed against dialysis buffer (20 mM Tris-HCl (pH 8), 250 mM NaCl). The sample was loaded onto a 1 ml HisTrap FF (GE Healthcare) column equilibrated with dialysis buffer to retain the TEV protease and the fraction of protein not digested. The column flowthrough was pooled, concentrated, and loaded onto a HiLoad Superdex 200 increase 10/300 GL (GE Healthcare) gel filtration column equilibrated in dialysis buffer. After size exclusion chromatography, the purest fractions judged by SDS-PAGE were pooled, concentrated at approximately 25 mg/ml and stored at −80˚C. Typical yields were 5 to 10 mg recombinant protein/L of culture medium.

## Peptide synthesis

Synthetic versions of the peptides were obtained from Proteogenix and Synpeptide. Quality check and >95% purity was confirmed by high-pressure liquid chromatography (HPLC).

## Peptide maturation assays

Overnight cultures of either *B. subtilis* strains 168, WB800, *Δmpr*, *Δbpr*, *ΔaprE*, *Δepr*, *ΔwprA*, *Δvpr*, *ΔnprB*, *ΔnprE* in GM1 media [41] were diluted 1:20 into GM1 media and incubated at 37˚C at 250 rpm. After 2 h, the OD$_{600}$ were measured and adjusted to 0.5 with GM1 media, and 1 ml of bacterial cultures were centrifuged at 8,000 g for 2 min at room temperature and the supernatants were filtered using 0,2 μm nitrocellulose filters; 1 μl of 10 mM synthetic

version of the mature pheromones (RRGHTA, RRGHTAS, SRGHTS, RGHTS, ERPVGT, Imat1[3T], Imat2[3T], Imat1[105], Imat2[105], and Imat3[105]) were added to 99 µl of supernatant obtaining a final concentration of 100 µm and incubated for 1 h at room temperature. After incubation, reaction was stopped by incubation at 95˚C for 5 min and sample was kept and used in Thermal Shift Assays (see before). In the kinetic assays, samples were inactivated at different times by incubation at 95˚C for 5 min. Experiments were validated by experimental triplicates of 3 biological replicates.

## Thermal shift assay

The thermal shift assay was conducted in a CFX Opus Real-Time PCR system (Biorad). Samples of 20 µl containing 20× Sypro Orange (Sigma-Aldrich) and 50 µm of protein in a 20 mM Tris (pH 8) and 250 mM NaCl buffer or filtered supernatant in peptide maturation assays (see below) were loaded in 96-well PCR plates. In the cases where peptide interaction was analyzed, synthetic peptides at a final concentration of 100 µm were added to the assays. Samples were heated from 20 to 85˚C in steps of 1 degree. Fluorescent intensity was plotted versus temperature and integrated with either CFX Opus Real-Time PCR automate model or GraphPad Prism software using a Boltzmann model to calculate melting temperatures.

## Microscale thermophoresis

Rap[3T] was fluorescently labeled with NT-647 amine reactive dye from the Monolith NT Protein Labeling kit RED-NHS (NanoTemper Technologies) according to the manufacturer's instructions. Briefly, 100 µm solution of the protein was mixed with 3× dye solution, incubated 30 min in dark, and then purified by gravity in a 5 ml size exclusion column equilibrated with dialysis buffer (20 mM Tris-HCl (pH 8), 250 mM NaCl) supplemented with 0.1% pluronic and 5% glycerol to remove free dye. Rap[105] was fluorescently labeled using His-Tag Labeling Kit RED-tris-NTA 2nd Generation. Briefly, 500 nM of protein was mixed with 5 µm dye in PBS supplemented with 0,02% of tween incubated in dark at room temperature for 30 min and centrifuged at 10,000 $g$ for 10 min at 4˚C, transferred to a fresh tube, and used in the assays immediately. Labeled proteins were used at a final optimized concentration of 20 nM in the case of Rap[3T] and 250 nM in the case of Rap[105]. For the measurements of EC$_{50}$, a 16-point 2-fold serial dilution (ranged from 2 to $3 \times 10^{-5}$ µm) of the peptides were prepared in presence of the corresponding labeled Rap protein. After 15-min incubation at room temperature, samples were filled into standard Capillaries (K005, NanoTemper Technologies) and MST measurements were performed on a Monolith NT.115 (NanoTemper Technologies) in RED channel using 60% to 75% of LED excitation power and medium or high MST power for Rap[3T] or Rap[105], respectively. The saturation curve with the peptides used as substrate showed clear sigmoidal behavior (S1 Fig) suggesting cooperative binding. Data analyses were performed with M.O. Affinity Analysis software (NanoTermper Technologies) applying cooperative (Hill coefficient) and non-cooperative (Kd model) models, with the first one yielding better fitting values.

## Size exclusion chromatography coupled with multi-angle light scattering (SEC-MALS)

In order to further analyze the oligomeric state of the Rap[3T] in presence and absence of different pheromones, SEC-MALS experiments were performed using a Shimadzu HPLC with a UV detector (Shimadzu, 280 nm) coupled with a MALS detector (TREOS II, Wyatt Technology), a dRI detector (Optilab T-rEX, Wyatt Technology) and a DynaPro NanoStar (Wyatt Technology). Size exclusion chromatography was performed with a column PROTEIN KW-

403 4F (Shodex), using a flow rate of 0.3 ml/min and a mobile phase consisting of 50 mM Hepes (pH 7) and 150 mM NaCl. For the calculation of the molecular weight in the presence of the pheromone, the mobile phase was supplemented with the pheromone under study at a final concentration of 10 μm. A volume of 20 μl of Rap$^{3T}$ sample at 1 mg/ml in the same buffer as the mobile phase was injected for each assay. In the case of analysis in the presence of pheromone, the Rap$^{3T}$ was preincubated for 10 min with 100 μm final concentration of the corresponding pheromone prior to injection. Data processing and molecular weight calculations were carried out using ASTRA 7.1.2 software (Wyatt Technology).

## Protein crystallization

The crystals were grown as sitting drops at 21˚C. Initial crystallization trials were set up in the Crystallogenesis service of the IBV-CSIC using commercial screens JBS I, II (JENA Biosciences) and MIDAS (Molecular Dimensions) in 96-well plates. Initial hits were reproduced and improved using the sitting drop method mixing equal volumes of protein at 10 mg/ml and homemade solutions. The Rap-Phr complexed were obtained in conditions with mother liquor with glycerol ethoxylate/tetrahydrofuran as a base. Specifically, the conditions were: Rap$^{3T}$-RRGHTAS and Rap$^{3T}$-RGHTS complexes crystallized in 25% glycerol ethoxylate, 0.2M NH$_4$Cl; Rap$^{3T}$-RRGHTA complex crystallized in 0.1M Tris (pH 8.5), 12.5% glycerol ethoxylate, 6% tetrahydrofuran; Rap$^{3T}$-SRGHTS complex crystallized in 0.1M Tris (pH 8.5), 15% glycerol ethoxylate, 4% tetrahydrofuran; and Rap$^{105}$-ERPVGT complex crystallized in 20% glycerol ethoxylate, 10% tetrahydrofuran, and 0.1M Tris-HCl (pH 8).

## Data collection, structure solution, and refinement

Prior to data collection, crystals were directly flash frozen in liquid nitrogen from crystallization drop or after cleaning in mother liquor. Diffraction data was collected from single crystals at 100˚K on ALBA (Barcelona, Spain) and DLS (Didcot, United Kingdom) synchrotrons. Data sets were processed with imosflm [42] and reduced using Aimless (CCP4) [43]. The data collection statistics for the best data sets used in structure determination are shown in S1 Table.

Phase for Rap$^{3T}$-peptide structures were determined by molecular replacement using Phaser (CCP4) [43]. For Rap$^{3T}$-RRGHTAS structure, the coordinates of RapF-PhrF complex (PDB entry: 4I9C) was used as search model. For the rest of Rap$^{3T}$ complexes, the Rap$^{3T}$-RRGHTAS coordinates were used as search model. The Rap$^{\phi105}$-ERPVGT structure was determined using an alphafold2 model [44] as search model. Following phase determination, model refinement was carried out combining manual building with Coot (v.0.9.8.8) [45] and computational refinement using phenix.refine [46]. Refinement statistics are summarized in S1 Table.

## Supporting information

**S1 Fig. Microscale thermophoresis experiments on Rap-pheromone interactions.** Dose-response slopes for the interaction between Rap$^{3T}$ and pheromone variants (RRGHTA, SRGHTS, RRGHTAS, RGHTS, and Imat1$^{3T}$) and Rap$^{105}$ with pheromone ERPVGT. Notice the sigmoidal and non-hyperbolic curve, indicating a cooperative effect on peptide binding. (TIFF)

**S2 Fig. Thermal shift slopes of peptide maturation assays.** Slopes of representative thermal shift assays with supernatants from *B. subtilis* 168, WB800, individual protease mutant strains and pheromone controls with Rap$^{3T}$ (left) and Rap$^{105}$ (right) for maturation of peptides

Imat1$^{3T}$(Rap$^{3T}$), Imat2$^{3T}$(Rap$^{3T}$), Imat1$^{105}$ (Rap$^{105}$), Imat2$^{105}$ (Rap$^{105}$), and Imat3$^{105}$ (Rap$^{105}$).
(TIFF)

**S3 Fig. Dimeric organization of Rap in crystal structures.** Cartoon representation of the dimers observed in the crystal structures of Rap$^{3T}$, Rap$^{105}$, RapJ (PDB 4GYO), RapH (PDB 3Q15), RapI (PDB 4I1A), and RapF (PDB 4I9E). All proteins show a similar dimerization mode interacting mainly with the TPR domains (orange-yellow) mainly through the C-terminal helix (in tones of red) with the 3HB domains (blue tones) facing outward from the dimer.
(TIFF)

**S4 Fig. All peptides induce identical conformation in Rap$^{3T}$.** (**a**) RMSD calculation from the superimposition of Cα atoms of Rap$^{3T}$ structures in complex with peptides SRGHTS, RGHTS, RRGHTA, and RRGHTAS. RMSD were calculated from individual monomers (lower) and dimers (upper). Number of atoms used in RMSD calculation is showed between parenthesis. (**b**) Superposition of 4 Rap$^{3T}$-Peptide complexes with different pheromone variants in monomeric state RRGHTAS (blue), RRGHTA (pink), RGHTS (light green), and SRGHTS (orange). (**c**) Superposition of dimers for the 4 Rap$^{3T}$ complexes with peptides RRGHTAS (blue), RRGHTA (pink), RGHTS (light green), and SRGHTS (orange).
(TIFF)

**S5 Fig. Rap$^{3T}$ is a dimer in solution.** Size exclusion chromatography multi-angle light scattering (SEC-MALS) chromatograms of Rap$^{3T}$ in absence (a) and presence of SRGHTS (b) and RRGHTA (c) pheromones. Chromatograms show the readings from the light scattering (dashed black line), refractive index (blue line), and ultraviolet (green line) detectors. The vertical axis represents the molecular mass. The horizontal red curves represent the calculated molecular masses. In all cases, the molecular weight calculated corresponds to a dimeric form (theoretical MW for the apo, SRGHTS and RRGHTA dimers are 90, 91.2, and 91.3 kDa, respectively).
(TIFF)

**S6 Fig. Rap$^{3T}$ peptide recognition.** Close view of the peptides (a) RRGHTA, (b) SRGHTS, (c) RRGHTAS, and (d) RGHTS bound to Rap$^{3T}$showing the peptides and the Rap$^{3T}$ interacting residues in sticks. The Rap$^{3T}$ structural elements where the recognition residues are placed are shown in translucent white cartoon. Signaling peptides and Rap$^{3T}$ interacting residues are labeled in red and black, respectively. In b, the Fo − Fc omit electron-density Fourier map contoured at 2σ and carved within 2.5 Å for RRGHTAS is shown in black.
(TIFF)

**S1 Table. Data collection and refinement statistics for Rap3T and Rap105 in complex with peptides.**
(XLSX)

**S2 Table. Intermolecular interactions in Rap$^{3T}$ and Rap$^{105}$ complexes with peptides.**
(XLSX)

**S3 Table. Pheromone variants present in Phr$^{3T}$ and Phr$^{105}$ families.**
(XLSX)

**S4 Table. Strains and plasmids used in this study.**
(XLSX)

**S5 Table. Oligonucleotides used in this study.**
(XLSX)

**S1 Data. The data underlying Fig 1B.**
(XLSX)

**S2 Data. The data underlying Fig 1C.**
(XLSX)

**S3 Data. The data underlying Figs 1D and S1.**
(XLSX)

**S4 Data. The data underlying Figs 2A–2C and S2.**
(XLSX)

**S5 Data. The data underlying Fig 2D.**
(XLSX)

**S6 Data. The data underlying Figs 2E–2H and S2.**
(XLSX)

**S7 Data. The data underlying Fig 5.**
(XLSX)

**S8 Data. The data underlying Fig 6.**
(XLSX)

**S9 Data. The data underlying S5 Fig.**
(XLSX)

## Acknowledgments

We thank Francisca Gallego-del Sol for her help collecting synchrotron data from crystal of Rap[105]-ERPVGT complex and ordering synthetic version of the gene *rap[105]*. We would like to thank the IBV-CSIC Crystallogenesis Facility for protein crystallization screenings. The structural results reported in this article derive from measurements made at the synchrotron DLS (Didcot, UK), ALBA (Cerdanyola del Valles, Spain), and ESRF (Grenoble, France). Data collection experiments for the best crystals were carried out at XALOC and I04 beamlines at ALBA and DLS Synchrotrons, respectively. X-ray diffraction data collection was supported by block allocation group (BAG) DLS Proposal MX28394, ALBA Proposal 2020074406, and ESRF proposal MX-2452. We acknowledge the ESRF, ALBA, and DLS synchrotrons for provision of beam time and we would like to thank beamline staff for assistance. Some figures in this manuscript have been created with Biorender.com.

## Author Contributions

**Conceptualization:** Alonso Felipe-Ruiz, José R. Penadés, Avigdor Eldar, Alberto Marina.

**Data curation:** Shira Omer Bendori, Avigdor Eldar, Alberto Marina.

**Formal analysis:** Alonso Felipe-Ruiz, Sara Zamora-Caballero, Shira Omer Bendori, José R. Penadés, Avigdor Eldar, Alberto Marina.

**Funding acquisition:** José R. Penadés, Avigdor Eldar, Alberto Marina.

**Investigation:** Alonso Felipe-Ruiz, Sara Zamora-Caballero, Shira Omer Bendori, Alberto Marina.

**Methodology:** Alonso Felipe-Ruiz.

**Project administration:** Avigdor Eldar.

**Resources:** Avigdor Eldar.

**Supervision:** Alberto Marina.

**Validation:** Sara Zamora-Caballero.

**Visualization:** Sara Zamora-Caballero, Shira Omer Bendori, José R. Penadés, Avigdor Eldar, Alberto Marina.

**Writing – original draft:** Alonso Felipe-Ruiz, José R. Penadés, Avigdor Eldar, Alberto Marina.

**Writing – review & editing:** Alonso Felipe-Ruiz, José R. Penadés, Alberto Marina.

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
