## [Editor Report · Decision Letter 0]

5 Apr 2024

Dear Dr Marina, 

Thank you for submitting your manuscript entitled "Maturation Mechanisms and Binding Dynamics of Duplicated Peptides Unlocks Extra Layers of Control in Bacterial Quorum Sensing" for consideration as a Research Article by PLOS Biology.

Your manuscript has now been evaluated by the PLOS Biology editorial staff, as well as by an academic editor with relevant expertise, and I am writing to let you know that we would like to send your submission out for external peer review.

Once your full submission is complete, your paper will undergo a series of checks in preparation for peer review. After your manuscript has passed the checks it will be sent out for review. To provide the metadata for your submission, please Login to Editorial Manager (https://www.editorialmanager.com/pbiology) within two working days, i.e. by Apr 07 2024 11:59PM.

Kind regards,

Melissa

Melissa Vazquez Hernandez, Ph.D.

Associate Editor

PLOS Biology

---

## [Decision Letter · Decision Letter 1]

13 May 2024

Dear Dr Marina,

Thank you for your patience while your manuscript "Maturation Mechanisms and Binding Dynamics of Duplicated Peptides Unlocks Extra Layers of Control in Bacterial Quorum Sensing" went through peer-review at PLOS Biology. Your manuscript has now been evaluated by the PLOS Biology editors, an Academic Editor with relevant expertise, and by three independent reviewers, being Sylvie Nessler reviewer #1.

As you will see in the reports, all reviewers are positive about the relevance of the work, but still some concerns should be addressed prior to publication. Reviewer #1 requires further clarification in some experiments and their interpretation, as well as the Imat13T and Imat23T results. Reviewer #2 would like you to define pseudopeptide. And Reviewer #3 asks that the reporters of the multiple protease KO strain are included.

IMPORTANT: While the complementation experiment suggested by reviewer #3 would provide additional rigor to the study, we do not consider this to be necessary for publication at PLOS Biology. Addressing all other concerns of the reviewers is, however, essential for further consideration of your manuscript for publication in PLOS Biology.

**IMPORTANT - SUBMITTING YOUR REVISION**

*Resubmission Checklist*

*Published Peer Review*

*PLOS Data Policy*

*Blot and Gel Data Policy*

Sincerely,

Melissa

Melissa Vazquez Hernandez, Ph.D.

Associate Editor

PLOS Biology

REVIEWERS' COMMENTS

Reviewer #1: 

The manuscript "Maturation Mechanisms and Binding Dynamics of Duplicated Peptides Unlocks Extra Layers of Control in Bacterial Quorum Sensing" by Felipe-Ruiz et al. aims to decipher the regulatory role of peptide pheromone duplication in the RRNPPA bacterial communication system. 

The authors use two Rap-Phr systems of bacteriophages phi105 and phi3T as models to investigate two types of duplications.

They combine in vitro interaction measurements with maturation assays using bacterial strains deleted of distinct protease genes and analyze their results in the light of structural data. They also analyze the conservation of these 2 types of pheromone duplication among other Rap-Phr systems. Overall, the experiments are well described and the results clearly illustrated.

The main result is the need, in both systems, for distinct specific proteases to produce the two active mature peptides. As each protease is expressed under specific metabolic conditions, they will produce distinct mature peptides in the case of divergent duplication, or distinct quantity of the same peptide in the case of conserved duplication, and will therefore regulate distinct biological processes (sporulation, biofilm formation..)

Altogether, this multidisciplinary study demonstrates that propeptide maturation plays an essential role in the regulation of these QS systems and that duplication of the mature peptide allows a single system to regulate distinct function depending on the availability of the secreted proteases. 

In view of the novelty of the results presented here, I recommend this manuscript for publication in PloS Biol.

However, the discussion concerning the cooperative binding of the peptide in both systems could be improved to support the regulatory mechanisms proposed for the two systems. In particular, the interpretation of the MST experiments and the calculation of the Hill coefficient could be better explained (Sup Fig 1 and Fig 1d). 

The authors consider that the reported cooperativity is in agreement with the dimeric organization of the Rap proteins but several papers (Baker and Neiditch 2011, Parashar et al 2013, Parashar et al 2011) showed that Rap proteins are monomeric in solution and most dimers observed in the crystal structures of Rap proteins are not considered as stable. 

In addition, to reinforce the main finding that distinct proteases are required to produce the two mature peptides, further details could be provided to explain the results of the thermal shift assays performed with culture supernatants of bacterial strains lacking each protease. In particular, the significance of the two slopes in Fig. 2c should be better explained and the result would be more convincing with all curves covering the same temperature range. Provide also the results presented in Fig. 2b and 2c for both Imat13T and Imat23T.

Some minor issues should also be considered:

- Missing references 

P4, line 65: add references for the conformational change induced upon peptide binding to RRNPPA

P4, lines 66-69: what is described here is only true for the Rap proteins but not for the RRNPPA transcription factors. Modulate the statement.

P5, line 102: Briefly introduce the arbitrium QS system of phage phi3T

P6, line 140: Rap dimerization is controversial. Discuss and add references.

P9, line 215: add a reference for the prototypical Rap dimer 

P16, line 407: add a reference for the neofunctionalization hypothesis

- Figures

In Sup Figure 1, explain what the x-axis represents. Dose = peptide concentration ? unit ? 

In Sup Table 1: add the CC1/2 values in the statistics for each data set and the Ramachandran data in the refinement statistics. Add a legend explaining Rmerge, Rwork, Rfree, unit of B-factors

The resolution of Figures 2c and 2d should be increased. 

Explain in the legend of Figure 2d how the relative maturation is calculated.

Sup Figure 3: Add a panel showing an omit map calculated with the complex of panel a) showing that electron density is missing for the terminal residue of the heptapeptide RRGHTAS. Discuss if there is enough space in the binding pocket to accommodate this additional C-terminal residue.

Reviewer #2: 

This manuscript very thoroughly investigates the function of duplicated pheromones in quorum sensing (QS) in Bacillus mobile genetic elements. This study involves the RRNPPA family of receptors and small unmodified peptide QS molecules. The first studied members of this system were the Rap/Phr that controlled sporulation and genetic competence in B. subtilis. Those Phr encoded a pre-peptide with the mature peptide encoded by the C-terminal pentapeptide. Since then, many more Rap/Phr sequence have been revealed, and many of the Phr genes encode more than one mature peptide. This had led to the hypothesis that the duplication was for the purpose of divergence in the function of the peptide. This study analyzes the production and functions of the duplicated peptides and shows that they have highly conserved function and are not under divergent selection. 

I just have one small suggestion which is to indicate how you are defining pseudopeptide on page 16. Some speculation as to the function of the pseudopeptide based on what you have learned in this study would also be useful. 

Reviewer #3: 

The manuscript entitled "Maturation Mechanisms and Binding Dynamics of Duplicated Peptides Unlocks Extra Layers of Control in Bacterial Quorum Sensing" is well-written and rationale presented. It explores the hypothesis that some cell-cell communication (CCC) peptides encode multiple active peptides within their sequence, that these are revealed by extracellular proteases, and in doing so provide an addition level of regulation for CCC. Further, these peptides are part of the widely distributed RRNPPA superfamily, such that the findings have broad implications.

Using genetic, biophysical and structural data, complemented by comparative genomics, the manuscript reveals a molecular mechanism for processing of CCC peptides from two Phr pre-peptides. They demonstrate that multiple active peptides are present with the pre-processed forms. They quantify their binding affinities to cognate receptors, capture the structure of the peptides and their receptors, identify processing proteases, and demonstrate phenotypic consequences of the binding. Thus, the paper is comprehensive in its level of analyses and the conclusion timely and relevant to bacterial population behaviors.

I have only a few minor comments and considerations.

(1) Please add statistics to 5. 

(2) Figure 2 suggests that the protease WprA is a major contributor to post-export processing of the peptides. However, I don't think the studies included a complemented strain. This would be highly recommended. Related, the addition of Imat13T and Imat23T to the phenotypes tested in Fig 5 is a very good addition. When the interpretation is combined with data presented in figure 2, it suggests that these peptides are processed by endogenous proteases. While not required, it would be rigorous to include the reporters in the multiple protease knockout strain, were the Phr peptide would be predicted to behave as the negative control. 

(3) The resolution of images 2C& 2D is low.

---

## [Editor Report · Decision Letter 2]

27 Jun 2024

Dear Dr Marina,

Thank you for your patience while we considered your revised manuscript "Maturation Mechanisms and Binding Dynamics of Duplicated Peptides Unlocks Extra Layers of Control in Bacterial Quorum Sensing" for publication as a Research Article at PLOS Biology. This revised version of your manuscript has been evaluated by the PLOS Biology editors, the Academic Editor.

Based on our Academic Editor's assessment of your revision, we are likely to accept this manuscript for publication. Please also make sure to address the following data and other policy-related requests.

a) We routinely suggest changes to titles to ensure maximum accessibility for a broad, non-specialist readership, and to ensure they reflect the contents of the paper. In this case, we would suggest a minor edit to the title, as follows. Please ensure you change both the manuscript file and the online submission system, as they need to match for final acceptance.

“Extracellular proteolysis of tandemly duplicated pheromone propeptides affords additional complexity to bacterial quorum sensing”

Please supply the numerical values either in the a supplementary file or as a permanent DOI’d deposition for the following figures:

Figure 1BCD, 2ABCDEFGH, 5AB, 6B, S1, S2, S5.

c) Please cite the location of the data clearly in all relevant main and supplementary Figure legends, e.g. “The data underlying this Figure can be found in S1 Data” or “The data underlying this Figure can be found in https://doi.org/10.5281/zenodo.XXXXX”

d) Please ensure that your Data Statement in the submission system accurately describes where your data can be found and is in final format, as it will be published as written there.

e) Per journal policy, if you have generated any custom code during the curse of this investigation, please make it available without restrictions upon publication. Please ensure that the code is sufficiently well documented and reusable, and that your Data Statement in the Editorial Manager submission system accurately describes where your code can be found.

f) Please note that per journal policy, the model system/species (Bacillus subtilis) studied should be clearly stated in the abstract of your manuscript.

We expect to receive your revised manuscript within two weeks. 

*Published Peer Review History*

*Press*

Sincerely,

Melissa

Melissa Vazquez Hernandez, Ph.D.

Associate Editor

PLOS Biology

---

## [Editor Report · Decision Letter 3]

9 Jul 2024

Dear Dr Marina,

Thank you for the submission of your revised Research Article "Extracellular proteolysis of tandemly duplicated pheromone  propeptides affords additional complexity to bacterial quorum sensing" for publication in PLOS Biology. On behalf of my colleagues and the Academic Editor, Ann M. Stock, I am pleased to say that we can in principle accept your manuscript for publication, provided you address any remaining formatting and reporting issues. These will be detailed in an email you should receive within 2-3 business days from our colleagues in the journal operations team; no action is required from you until then. Please note that we will not be able to formally accept your manuscript and schedule it for publication until you have completed any requested changes.

IMPORTANT: Thank you for attending to the previous editorial requests. However, before publication, two remaining requests must be addressed. The title from the suplementary data file 9 is mislabeled and should state "The data underlying Figure S5". Additionally, please provide the references mentioned for Supplementary Table 4, in the main text under Materials and methods sections in the corresponding subheadings "Bacterial strains" (the first two references) and "plasmids and cloning" (last reference); this should not affect the number of references provided. I have asked my colleagues to include this request alongside their own.

PRESS

Sincerely, 

Melissa Vazquez Hernandez, Ph.D.

Associate Editor

PLOS Biology
